# CARPE DIEM, SEIZE THE SAMPLES UNCERTAIN "AT THE MOMENT" FOR ADAPTIVE BATCH SELECTION

## ABSTRACT

The performance of deep neural networks is significantly affected by how well mini-batches are constructed. In this paper, we propose a novel adaptive batch selection algorithm called **Recency Bias** that exploits the uncertain samples predicted inconsistently *in recent iterations*. The historical label predictions of each sample are used to evaluate its predictive uncertainty within a *sliding window*. By taking advantage of this design, *Recency Bias* not only accelerates the training step but also achieves a more accurate network. We demonstrate the superiority of *Recency Bias* by extensive evaluation on two independent tasks. Compared with existing batch selection methods, the results showed that *Recency Bias* reduced the test error by up to $20.5\%$ in a fixed wall-clock training time. At the same time, it improved the training time by up to $59.3\%$ to reach the same test error.

## 1 INTRODUCTION

Stochastic gradient descent (SGD) for *randomly* selected mini-batch samples is commonly used to train deep neural networks (DNNs). However, many recent studies have pointed out that the performance of DNNs is heavily dependent on how well the mini-batch samples are selected (Shrivastava et al., 2016; Chang et al., 2017; Katharopoulos & Fleuret, 2018). In earlier approaches, a sample's *difficulty* is employed to identify proper mini-batch samples, and these approaches achieve a more accurate and robust network (Han et al., 2018) or expedite the training convergence of SGD (Loshchilov & Hutter, 2016). However, the two opposing difficulty-based strategies, i.e., preferring *easy* samples (Kumar et al., 2010; Han et al., 2018) versus *hard* samples (Loshchilov & Hutter, 2016; Shrivastava et al., 2016), work well in different situations. Thus, for practical reasons to cover more diverse situations, recent approaches begin to exploit a sample's *uncertainty* that indicates the consistency of previous predictions (Chang et al., 2017; Song et al., 2019).

An important question here is how to evaluate the sample's uncertainty based on its historical predictions during the training process. Intuitively, because a series of historical predictions can be seen as a series of data indexed in chronological order, the uncertainty can be measured based on *two* forms of handling time-series observations: *(i)* a *growing window* (Figure 1(a)) that consistently increases the size of a window to use all available observations and *(ii)* a *sliding window* (Figure 1(b)) that maintains a window of a fixed size on the most recent observations by deleting outdated ones. While the state-of-the-art algorithm, *Active Bias* (Chang et al., 2017), adopts the growing window, we propose to use the sliding window in this paper.

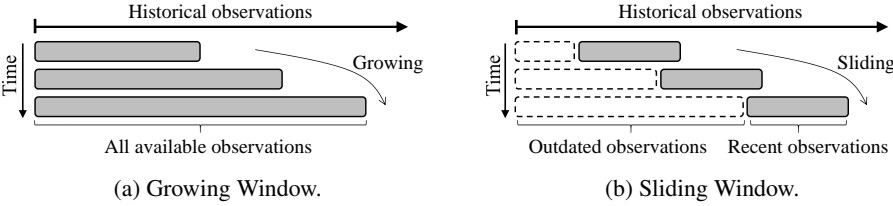

(a) Growing Window.  (b) Sliding Window.

Figure 1: Two forms of handling the time-series observations.

In more detail, *Active Bias* recognizes uncertain samples based on the inconsistency of the predictions in the *entire* history of past SGD iterations. Then, it emphasizes such uncertain samples by choosing them with high probability for the next mini-batch. However, according to our experiments presented

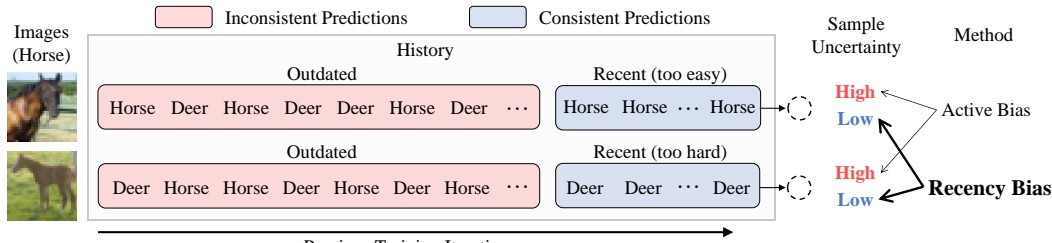

Figure 2: The difference in sample uncertainty estimated by *Active Bias* and *Recency Bias*.

in Section 5.2, such uncertain samples *slowed down* the convergence speed of training, though they ultimately reduced the generalization error. This weakness is attributed to the inherent limitation of the growing window, where older observations could be too outdated (Torgo, 2011). In other words, the outdated predictions no longer represent a network's current behavior. As illustrated in Figure 2, when the label predictions of two samples were inconsistent for a long time, *Active Bias* invariably regards them as highly uncertain, although their recent label predictions become consistent along with the network's training progress. This characteristic evidently entails the risk of emphasizing uninformative samples that are too easy or too hard at the current moment, thereby slowing down the convergence speed of training.

Therefore, we propose a simple but effective batch selection method, called **Recency Bias**, that takes advantage of the *sliding window* to evaluate the uncertainty in *fresher* observations. As opposed to *Active Bias*, *Recency Bias* excludes the outdated predictions by managing a sliding window of a fixed size and picks up the samples predicted inconsistently within the sliding window. Thus, as shown in Figure 2, the two samples uninformative at the moment are no longer selected by *Recency Bias* simply because their recent predictions are consistent. Consequently, since informative samples are effectively selected throughout the training process, this strategy not only accelerates the training speed but also leads to a more accurate network.

To validate the superiority of *Recency Bias*, two popular convolutional neural networks (CNNs) were trained for two independent tasks: image classification and fine tuning. We compared *Recency Bias* with not only random batch selection (baseline) but also two state-of-the-art batch selection strategies. Compared with three batch selection strategies, *Recency Bias* provided a relative reduction of test error by $1.81\%$–$20.5\%$ in a fixed wall-clock training time. At the same time, it significantly reduced the execution time by $24.6\%$–$59.3\%$ to reach the same test error.

## 2 RELATED WORK

Let $\mathcal{D} = \{(x_i, y_i)|1 \leq i \leq N\}$ be the entire training dataset composed of a sample $x_i$ with its true label $y_i$, where $N$ is the total number of training samples. Then, a straightforward strategy to construct a mini-batch $\mathcal{M} = \{(x_i, y_i)|1 \leq i \leq b\}$ is to select $b$ samples *uniformly at random* (i.e., $P(x_i|\mathcal{D}) = 1/N$) from the training dataset $\mathcal{D}$.

Because not all samples have an equal impact on training, many research efforts have been devoted to develop advanced *sampling schemes*. Bengio et al. (2009) first took easy samples and then gradually increased the difficulty of samples using heuristic rules. Kumar et al. (2010) determined the easiness of the samples using their prediction errors. Recently, Tsvetkov et al. (2016) used Bayesian optimization to learn an optimal curriculum for training dense, distributed word representations. Sachan & Xing (2016) emphasized that the right curriculum must introduce a small number of the samples dissimilar to those previously seen. Fan et al. (2017) proposed a neural data filter based on reinforcement learning to select training samples adaptively. However, it is common for deep learning to emphasize *hard* samples because of the plethora of easy ones (Katharopoulos & Fleuret, 2018).

Loshchilov & Hutter (2016) proposed a *difficulty*-based sampling scheme, called *Online Batch*, that uses the rank of the loss computed from previous epochs. *Online Batch* sorts the previously computed losses of samples in descending order and exponentially decays the sampling probability of a sample according to its rank $r$. Then, the $r$-th ranked sample $x(r)$ is selected with the probability dropping by a factor of $\exp\big(\log(s_e)/N\big)$, where $s_e$ is the *selection pressure* parameter that affects the probability gap between the most and the least important samples. When normalized to sum to 1.0, the probability $P(x(r)|\mathcal{D}; s_e)$ is defined by Eq. (1). It has been reported that *Online Batch*

accelerates the convergence of training but deteriorates the generalization error because of the overfitting to hard training samples (Loshchilov & Hutter, 2016).

$$P(x(r)|\mathcal{D}; s_e) = \frac{1/\exp\big(\log(s_e)/N\big)^r}{\sum_{j=1}^{N} 1/\exp\big(\log(s_e)/N\big)^j} \tag{1}$$

Most close to our work, Chang et al. (2017) devised an *uncertainty*-based sampling scheme, called *Active Bias*, that chooses uncertain samples with high probability for the next batch. *Active Bias* maintains the history $\mathcal{H}_i^{t-1}$ that stores *all* $h(y_i|x_i)$ before the current iteration $t$ (i.e., growing window), where $h(y_i|x_i)$ is the softmax probability of a given sample $x_i$ for its true label $y_i$. Then, it measures the uncertainty of the sample $x_i$ by computing the variance over *all* $h(y_i|x_i)$ in $\mathcal{H}_i^{t-1}$ and draws the next mini-batch samples based on the normalized probability $P(x_i|\mathcal{D}, \mathcal{H}_i^{t-1}; \epsilon)$ in Eq. (2), where $\epsilon$ is the smoothness constant to prevent the low variance samples from never being selected again. As mentioned earlier in Section 1, *Active Bias* slows down the training process because the oldest part in the history $\mathcal{H}_i^{t-1}$ no longer represents the current behavior of the network.

$$P(x_i|\mathcal{D}, \mathcal{H}_i^{t-1}; \epsilon) = \frac{\hat{std}(\mathcal{H}_i^{t-1}) + \epsilon}{\sum_{j=1}^{N} \big(\hat{std}(\mathcal{H}_j^{t-1}) + \epsilon\big)}, \ \hat{std}(\mathcal{H}_i^{t-1}) = \sqrt{var\big(h(y_i|x_i)\big) + \frac{var\big(h(y_i|x_i)\big)^2}{|\mathcal{H}_i^{t-1}|}} \tag{2}$$

For the completeness of the survey, we include the recent studies on submodular batch selection. Joseph et al. (2019) and Wang et al. (2019) designed their own submodular objectives that cover diverse aspects, such as sample redundancy and sample representativeness, for more effective batch selection. Differently from their work, we explore the issue of truly uncertain samples in an orthogonal perspective. Our uncertainty measure can be easily injected into their submodular optimization framework as a measure of sample informativeness.

In Section 5, we will confirm that *Recency Bias* outperforms *Online Batch* and *Active Bias*, which are regarded as two state-of-the-art adaptive batch selection methods for deep learning.

## 3 *Recency Bias* COMPONENTS

### 3.1 CRITERION OF AN UNCERTAIN SAMPLE

The main challenge of *Recency Bias* is to identify the samples whose *recent* label predictions are highly inconsistent, which are neither too easy nor too hard at the moment. Thus, we adopt the *predictive uncertainty* (Song et al., 2019) in Definition 3.1 that uses the information entropy (Chandler, 1987) to measure the inconsistency of recent label predictions. Here, the sample with high predictive uncertainty is regarded as uncertain and selected with high probability for the next mini-batch.

**Definition 3.1. (Predictive Uncertainty)** Let $\hat{y}_{it} = \Phi(x_i, \theta_t)$ be the predicted label of a sample $x_i$ at time $t$ and $\mathcal{H}_{x_i}(q) = \{\hat{y}_{t_1}, \hat{y}_{t_2}, \ldots, \hat{y}_{t_q}\}$ be the label history of the sample $x_i$ that stores the predicted labels at the previous $q$ times, where $\Phi$ is a neural network. The label history $\mathcal{H}_{x_i}(q)$ corresponds to the *sliding window* of size $q$ to compute the uncertainty of the sample $x_i$. Next, $p(y_i|x_i; q)$ is formulated such that it provides the probability of the label $y_i \in \{1, 2, ..., k\}$ estimated as the label of the sample $x_i$ based on $\mathcal{H}_{x_i}(q)$ as in Eq. (3), where $[\cdot]$ is the Iverson bracket[1].

$$p(y_i|x_i; q) = \frac{\sum_{\hat{y}_i \in \mathcal{H}_{x_i}(q)} [\hat{y}_i = y_i]}{|\mathcal{H}_{x_i}(q)|} \tag{3}$$

Then, to quantify the uncertainty of the sample $x_i$, the *predictive uncertainty* $F(x_i; q)$ is defined using the empirical entropy as in Eq. (4). Because the uncertainty is bounded, we add the standardization term $\delta$ to normalize the value to $[0, 1]$. For $k$ classes, $\delta$ is the maximum entropy when $\forall_j p(j|x_i; q) = 1/k$.

$$F(x_i; q) = -(1/\delta) \sum_{j=1}^{k} p(j|x_i; q) \log p(j|x_i; q) \tag{4}$$

$$\delta = -\log(1/k) \ \square$$

---

[1] The Iverson bracket $[p]$ returns 1 if $p$ is true; 0 otherwise.

## 3.2 Sampling Probability for Mini-batch Construction

To construct next mini-batch samples, we assign the sampling probability according to the predictive uncertainty in Definition 3.1. Motivated by Loshchilov & Hutter (2016), the sampling probability of a given sample $x_i$ is exponentially decayed with its predictive uncertainty $F(x_i; q)$. In detail, we adopt the quantization method (Chen & Wornell, 2001) and use the quantization index to decay the sampling probability. The index is obtained by the simple quantizer $Q$ in Eq. (5), where $\Delta$ is the quantization step size. Compared with the rank-based index (Loshchilov & Hutter, 2016), the quantization index is known to well reflect the difference in actual values (Widrow et al., 1996).

$$Q\big(F(x_i; q)\big) = \lceil \big(1 - F(x_i; q)\big)/\Delta \rceil, \ \ 0 \le F(x_i; q) \le 1 \tag{5}$$

In Eq. (5), we set $\Delta$ to be $1/N$ such that the index is bounded to $N$ (the total number of samples). Then, the sampling probability $P(x_i | \mathcal{D}; s_e)$ is defined as in Eq. (6). The higher the predictive uncertainty, the smaller the quantization index. Therefore, a higher sampling probability is assigned for uncertain samples in Eq. (6).

$$P(x_i | \mathcal{D}; s_e) = \frac{1/\exp\big(\log(s_e)/N\big)^{Q(F(x_i;q))}}{\sum_{j=1}^{N} 1/\exp\big(\log(s_e)/N\big)^{Q(F(x_j;q))}} \tag{6}$$

Meanwhile, it is known that using only some part of training data exacerbates the overfitting problem at a late stage of training (Loshchilov & Hutter, 2016; Zhou & Bilmes, 2018). Thus, to alleviate the problem, we include more training samples as the training progresses by exponentially decaying the selection pressure $s_e$ as in Eq. (7). At each epoch $e$ from $e_0$ to $e_{end}$, the selection pressure $s_e$ exponentially decreases from $s_{e_0}$ to 1. Because this technique gradually reduces the sampling probability gap between the most and the least uncertain samples, more diverse samples are selected for the next mini-batch at a later epoch. When the selection pressure $s_e$ becomes 1, the mini-batch samples are randomly chosen from the entire dataset.

$$s_e = s_{e_0} \Big( \exp\big( \log\left(1/s_{e_0}\right)/(e_{end} - e_0) \big) \Big)^{e - e_0} \tag{7}$$

## 4  *Recency Bias* Algorithm

---
**Algorithm 1** *Recency Bias* Algorithm

---
INPUT: $\mathcal{D}$: data, $epochs$, $b$: batch size, $q$: window size, $s_{e_0}$: initial selection pressure, $\gamma$: warm-up
OUTPUT: $\theta_t$: model parameter
1: $t \leftarrow 1$;
2: $\theta_t \leftarrow$ Initialize the model parameter;
3: **for** $i = 1$ **to** $epochs$ **do**
4:   /* **Sampling Probability Derivation** */
5:   **if** $i > \gamma$ **then**
6:     $s_e \leftarrow$ Decay_Selection_Pressure($s_{e_0}$, $i$);   /* Decaying $s_e$ by Eq. (7) */
7:     **for** $m = 1$ **to** $N$ **do**    /* Updating the index and the sampling probability in a batch */
8:       $q\_dict[x_m] = Q\big(F(x_m; q)\big)$;        /* By Eq. (5) */
9:     $p\_table \leftarrow$ Compute_Prob($q\_dict$, $s_e$); /* By Eq. (6) */
10:   /* **Network Training** */
11:   **for** $j = 1$ **to** $N/b$ **do**   /* Mini-batch */
12:     **if** $i \le \gamma$ **then**   /* Warm-up */
13:       $\{(x_1, y_1), \ldots, (x_b, y_b)\} \leftarrow$ Randomly select next mini-batch samples;
14:     **else** /* Adaptive batch selection */
15:       $\{(x_1, y_1), \ldots, (x_b, y_b)\} \leftarrow$ Select next mini-batch samples based on $p\_table$;
16:     $losses, labels \leftarrow$ Inference_Step($\{(x_1, y_1), \ldots, (x_b, y_b)\}$, $\theta_t$);   /* Forward */
17:     $\theta_{t+1} \leftarrow$ SGD_Step($losses$, $\theta_t$);   /* Backward */
18:     Update_Label_History($labels$);   /* By Definition 3.1 */
19:     $t \leftarrow t + 1$;
20: **return** $\theta_t$;

---

Algorithm 1 describes the overall procedure of *Recency Bias*. The algorithm requires a warm-up period of $\gamma$ epochs because the quantization index for each sample is not confirmed yet. During the warm-up period, which should be at least $q$ epochs ($\gamma \ge q$) to obtain the label history of size

$q$, randomly selected mini-batch samples are used for the network update (Lines 12–13). After the warm-up period, the algorithm decays the selection pressure $s_e$ and updates not only the quantization index but also the sampling probability in a batch at the beginning of each epoch (Lines 4–9). Subsequently, the uncertain samples are selected for the next mini-batch according to the updated sampling probability (Line 14–15), and then the label history is updated along with the network update (Lines 16–19).

Overall, the key technical novelty of *Recency Bias* is to incorporate the notion of a *sliding window* (Line 8) rather than a growing window into adaptive batch selection, thereby improving both training speed and generalization error.

**Time Complexity**: The main "additional" cost of *Recency Bias* is the derivation of the sampling probability for each sample (Lines 4–9). Because only simple mathematical operations are needed per sample, its time complexity is linear to the number of samples (i.e., $O(N)$), which is negligible compared with that of the forward and backward steps of a complex network (Lines 16–17). Therefore, we contend that *Recency Bias* does *not* add the complexity of an underlying optimization algorithm.

## 5 EVALUATION

We empirically show the improvement of *Recency Bias* over not only *Random Batch* (baseline) but also *Online Batch* (Loshchilov & Hutter, 2016) and *Active Bias* (Chang et al., 2017), which are two state-of-the-art adaptive batch selections. In particular, we elaborate on the effect of the sliding window approach (*Recency Bias*) compared with the growing window approach (*Active Bias*). *Random Batch* selects next mini-batch samples uniformly at random from the entire dataset. *Online Batch* selects hard samples based on the rank of the loss computed from previous epochs. *Active Bias* selects uncertain samples with high variance of true label probabilities in the growing window. All the algorithms were implemented using TensorFlow 1.8.0 and executed using a single NVIDIA Titan Volta GPU. For reproducibility, we provide the source code at `https://github.com/anonymized`.

Image classification and fine-tuning tasks were performed to validate the superiority of *Recency Bias*. Because fine-tuning is used to quickly adapt to a new dataset, it is suitable to reap the benefit of fast training speed. In support of reliable evaluation, we repeated every task *thrice* and reported the average and standard error of the best test errors. The *best test error* in a given time has been widely used for the studies on fast and accurate training (Katharopoulos & Fleuret, 2018; Loshchilov & Hutter, 2016).

### 5.1 ANALYSIS ON SELECTED MINI-BATCH SAMPLES

For an in-depth analysis on selected samples, we plot the loss distribution of mini-batch samples selected from CIFAR-10 by four different strategies in Figure 3. *(i)* The distribution of *Online Batch* is the most skewed toward high loss by the design principle of selecting hard samples. *(ii)* *Active Bias* emphasizes *moderately hard* samples at an early training stage in considering that its loss distribution lies between those of *Random Batch* and *Online Batch*. However, owing to the outdated predictions caused by the growing window, the proportion of easy samples with low loss increases at a late training stage. These easy samples, which are misclassified as uncertain at that stage, tend to make the convergence of training slow down. *(iii)* In contrast to *Active Bias*, by virtue of the sliding window, the distribution of *Recency Bias* lies between those of *Random Batch* and *Online Batch* regardless of the training stage. Consequently, *Recency Bias* continues to highlight the moderately hard samples, which are likely to be informative, during the training process.

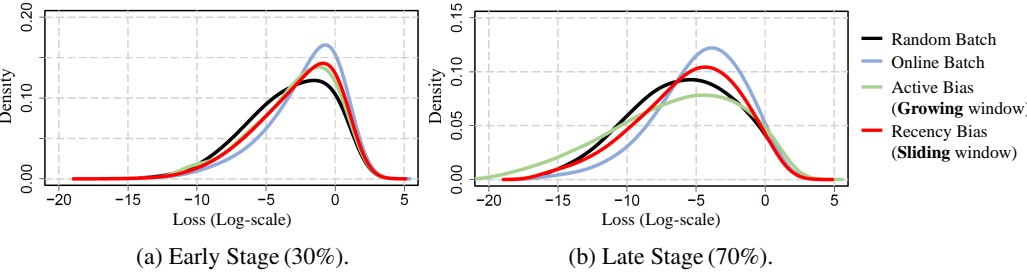

(a) Early Stage (30%).      (b) Late Stage (70%).

Figure 3: The loss distribution of mini-batch samples selected by four batch selection strategies: (a) and (b) show the loss distribution at the $30\%$ and $70\%$ of total training epochs, respectively.

### 5.2 TASK I: IMAGE CLASSIFICATION

**Experiment Setting**: We trained DenseNet (L=40, k=12) and ResNet (L=50) with a momentum optimizer and an SGD optimizer on *three* benchmark datasets: MNIST (10 classes)[2], classification of handwritten digits (LeCun, 1998), and CIFAR-10 (10 classes)[3] and CIFAR-100 (100 classes)[3], classification of a subset of 80 million categorical images (Krizhevsky et al., 2014). Specifically, we used data augmentation, batch normalization, a momentum of 0.9, and a batch size of 128. As for the algorithm parameters, we fixed the window size $q = 10$ and the initial selection pressure $s_{e_0} = 100$,[4] which were the best values found by the grid search (see Appendix A for details). The warm-up epoch $\gamma$ was set to be 15. To reduce the performance variance caused by randomly initialized model parameters, all parameters were shared by all algorithms during the warm-up period. Regarding the training schedule, we trained the network for $40,000$ iterations and used an initial learning rate of 0.1, which was divided by 10 at $50\%$ and $75\%$ of the total number of training iterations.

**Results**: Figure 4 shows the convergence curves of training loss and test error for four batch selection strategies using DenseNet and a momentum optimizer. In order to highlight the improvement of *Recency Bias* over the baseline (*Random Batch*), their lines are dark colored. The best test errors in Figures 4(b), 4(d), and 4(f) are summarized on the left side of Table 1.

In general, *Recency Bias* achieved the most accurate network while accelerating the training process on all datasets. The training loss of *Recency Bias* converged faster (Figures 4(a), 4(c), and 4(e)) without the increase in the generalization error, thereby achieving the lower test error (Figures 4(b), 4(d), and 4(f)). In contrast, the test error of *Online Batch* was not the best even if its training loss converged the fastest among all strategies. As the training difficulty increased from CIFAR-10 to CIFAR-100, the test error of *Online Batch* became even worse than that of *Random Batch*. That is, emphasizing hard samples accelerated the training step but made the network overfit to hard samples. Meanwhile, *Active Bias* was prone to make the network better generalized on test data. In CIFAR-10, despite its highest training loss, the test error of *Active Bias* was better than that of *Random Batch*. However, *Active Bias* slowed down the training process because of the limitation of growing windows, as discussed in Section 5.1. We note that, although both *Recency Bias* and *Active Bias* exploited uncertain samples, only *Recency Bias* based on sliding windows succeeded to not only speed up the training process but also reduce the generalization error.

The results of the best test error for ResNet or an SGD optimizer are summarized in Tables 1 and 2 (see Appendix C for more details). Regardless of a neural network and an optimizer, *Recency Bias* achieved the lowest test error except in MNIST with an SGD optimizer. The improvement of *Recency Bias* over the others was higher with an SGD optimizer than with a momentum optimizer.

Table 1: The best test errors (%) of four batch selection strategies using **DenseNet**.

| Optimizer | Momentum in Figure 4 | | | SGD in Figure 9 (Appendix C.1) | | |
|---|---|---|---|---|---|---|
| Method | MNIST | CIFAR-10 | CIFAR-100 | MNIST | CIFAR-10 | CIFAR-100 |
| *Random Batch* | $0.527 \pm 0.03$ | $7.33 \pm 0.09$ | $28.0 \pm 0.16$ | $1.23 \pm 0.03$ | $14.9 \pm 0.09$ | $40.2 \pm 0.06$ |
| *Online Batch* | $0.514 \pm 0.01$ | $7.00 \pm 0.10$ | $28.4 \pm 0.25$ | $0.765 \pm 0.02$ | $13.5 \pm 0.02$ | $40.7 \pm 0.12$ |
| *Active Bias* | $0.616 \pm 0.03$ | $7.07 \pm 0.04$ | $27.9 \pm 0.11$ | $\mathbf{0.679 \pm 0.02}$ | $14.2 \pm 0.25$ | $42.9 \pm 0.05$ |
| *Recency Bias* | $\mathbf{0.490 \pm 0.02}$ | $\mathbf{6.60 \pm 0.02}$ | $\mathbf{27.1 \pm 0.19}$ | $0.986 \pm 0.06$ | $\mathbf{13.2 \pm 0.11}$ | $\mathbf{38.7 \pm 0.11}$ |

Table 2: The best test errors (%) of four batch selection strategies using **ResNet**.

| Optimizer | Momentum in Figure 10 (Appendix C.2) | | | SGD in Figure 11 (Appendix C.3) | | |
|---|---|---|---|---|---|---|
| Method | MNIST | CIFAR-10 | CIFAR-100 | MNIST | CIFAR-10 | CIFAR-100 |
| *Random Batch* | $0.636 \pm 0.04$ | $10.2 \pm 0.12$ | $33.2 \pm 0.07$ | $1.16 \pm 0.03$ | $12.7 \pm 0.09$ | $40.1 \pm 0.16$ |
| *Online Batch* | $0.666 \pm 0.05$ | $10.1 \pm 0.05$ | $33.4 \pm 0.01$ | $0.890 \pm 0.03$ | $12.2 \pm 0.08$ | $40.7 \pm 0.09$ |
| *Active Bias* | $0.613 \pm 0.04$ | $10.6 \pm 0.08$ | $34.2 \pm 0.07$ | $\mathbf{0.804 \pm 0.01}$ | $13.5 \pm 0.07$ | $45.6 \pm 0.07$ |
| *Recency Bias* | $\mathbf{0.607 \pm 0.01}$ | $\mathbf{9.79 \pm 0.04}$ | $\mathbf{32.4 \pm 0.04}$ | $0.972 \pm 0.03$ | $\mathbf{11.6 \pm 0.09}$ | $\mathbf{38.9 \pm 0.14}$ |

---

[2]http://yann.lecun.com/exdb/mnist

[3]https://www.cs.toronto.edu/~kriz/cifar.html

[4]*Online Batch* also used the same decaying selection pressure value.

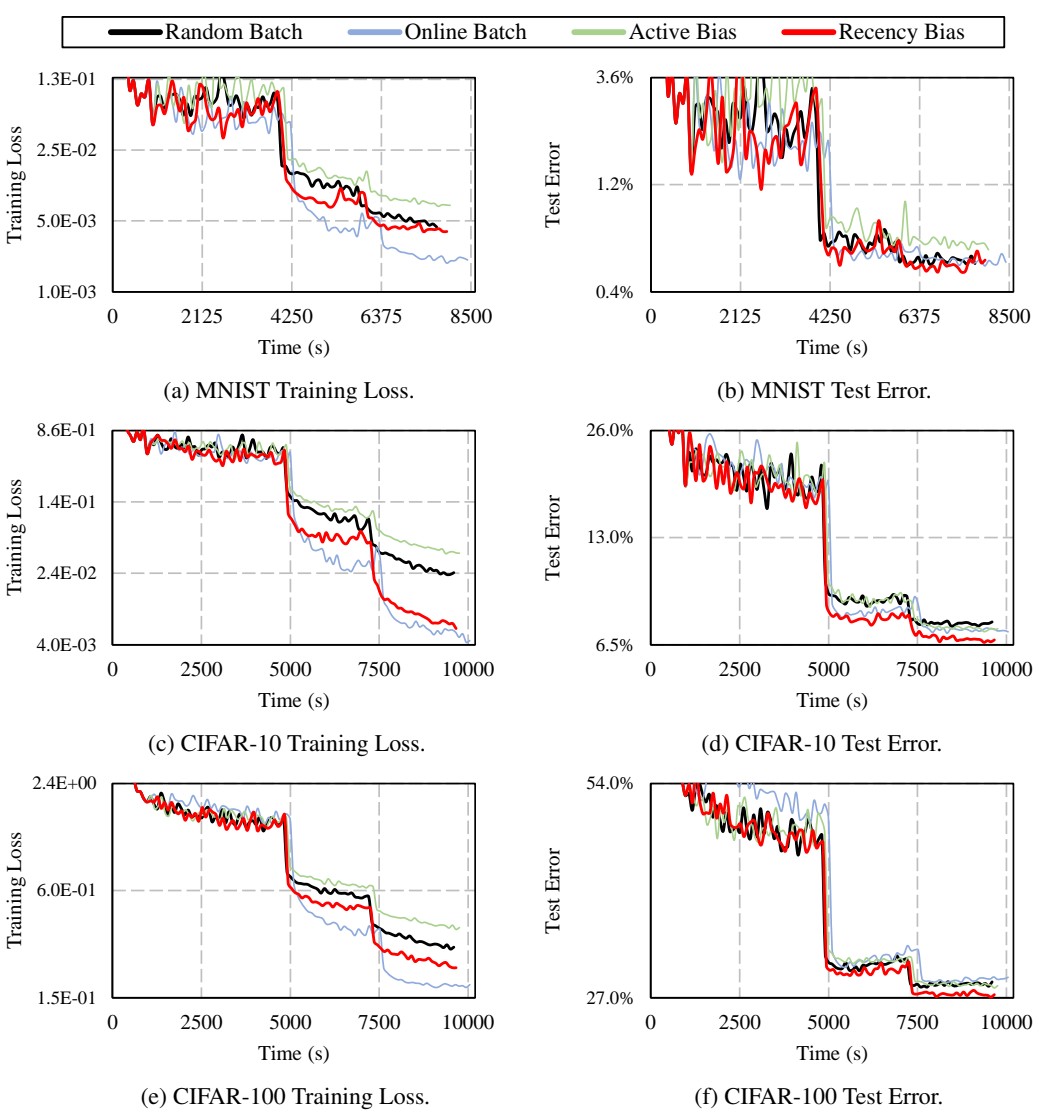

Figure 4: Convergence curves of four batch selection strategies using **DenseNet with momentum**.

## 5.3 TASK II: FINE-TUNING

**Experiment Setting**: We prepared DenseNet (L=121, k=32) previously trained on ImageNet (Deng et al., 2009) and then fine-tuned the network on *two* benchmark datasets: MIT-67 (67 classes)[5], classification of indoor scenes (Quattoni & Torralba, 2009), and Food-100 (100 classes)[6], classification of popular foods in Japan (Kawano & Yanai, 2014). After replacing the last classification layer, the network was trained end-to-end for 50 epochs with a batch size 32 and a constant learning rate $2 \times 10^{-4}$. Data augmentation was not applied here. The other configurations were the same as those in Section 5.2.

**Results on Test Error**: Figure 5 shows the convergence curves of training loss and test error for the fine-tuning task on MIT-67 and Food-100. Overall, all convergence curves showed similar trends to those of the classification task in Figure 4. Only *Recency Bias* converged faster than *Random Batch* in both training loss and test error. *Online Batch* converged the fastest in training loss, but its test error was rather higher than *Random Batch* owing to the overfitting. *Active Bias* converged the

---

[5] http://web.mit.edu/torralba/www/indoor.html
[6] http://foodcam.mobi/dataset100.html

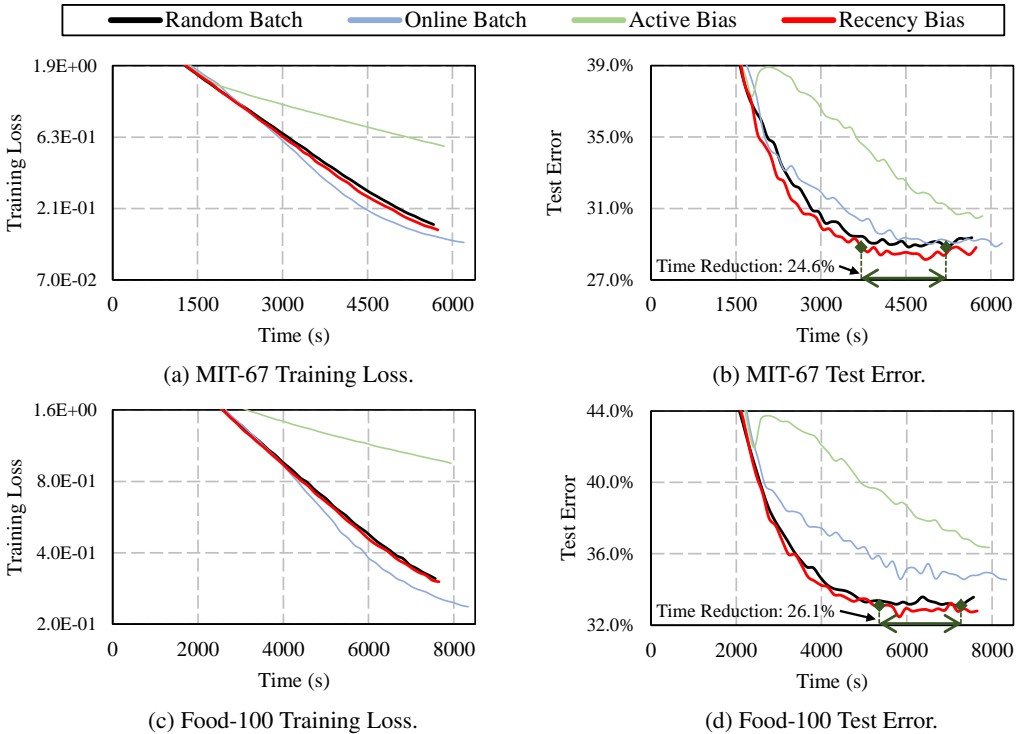

Figure 5: Convergence curves for fine-tuning on two benchmark datasets.

Table 3: **Recency Bias**'s reduction in training time over other batch selection strategies.

| Method | MIT-67 | FOOD-100 |
|---|---|---|
| *Random Batch* | $(5,218 - 3,936)/5,218 \times 100 = \mathbf{24.6\%}$ | $(7,263 - 5,365)/7,263 \times 100 = \mathbf{26.1\%}$ |
| *Online Batch* | $(6,079 - 3,823)/6,079 \times 100 = \mathbf{37.1\%}$ | $(8,333 - 3,685)/8,333 \times 100 = \mathbf{55.8\%}$ |
| *Active Bias* | $(5,738 - 3,032)/5,738 \times 100 = \mathbf{47.2\%}$ | $(7,933 - 3,227)/7,933 \times 100 = \mathbf{59.3\%}$ |

slowest in both training loss and test error. Quantitatively, compared with *Random Batch*, *Recency Bias* reduced the test error by $2.88\%$ and $1.81\%$ in MIT-67 and Food-100, respectively.

**Results on Training Time**: Moreover, to assess the performance gain in training time, we computed the reduction in the training time taken to reach the same error. For example, in Figure 5(b), the best test error of $28.8\%$ achieved in $5,218$ seconds by *Random Batch* could be achieved only in $3,936$ seconds by *Recency Bias*; thus, *Recency Bias* improved the training time by $24.6\%$. Table 3 summarizes the reduction in the training time of *Recency Bias* over three other batch selection strategies. Notably, *Recency Bias* improved the training time by $24.6\%$–$47.2\%$ and $26.1\%$–$59.3\%$ in fine-tuning MIT-67 and FOOD-100 datasets, respectively.

## 5.4 ABLATION STUDY ON SELECTION PRESSURE

For an ablation study on the selection pressure, we trained DenseNet (L=40, k=12) on two benchmark datasets using *Recency Bias* with *four* different decaying strategies: $s_e : 10 \to 10$, $s_e : 100 \to 100$, $s_e : 10 \to 1$, and $s_e : 100 \to 1$. The first two strategies used different initial selection pressures without decaying, but the remaining strategies exponentially decayed their initial selection pressures to 1. We used a momentum optimizer and the other experimental configurations were the same as those in Section 5.2.

Figure 6 shows the convergence curves of *Recency Bias* using the different decaying strategies along with that of *Random Batch*. Generally, the two strategies without decaying (i.e., $s_e : 10 \to 10$, $s_e : 100 \to 100$) showed much faster convergence speed in training loss compared with those with decaying (i.e., $s_e : 10 \to 1$, $s_e : 100 \to 1$). However, as mentioned earlier in Section 3.2, the two strategies without decaying exacerbated the overfitting problem because they only used the training samples classified as highly uncertain. Accordingly, their test errors were rather higher than

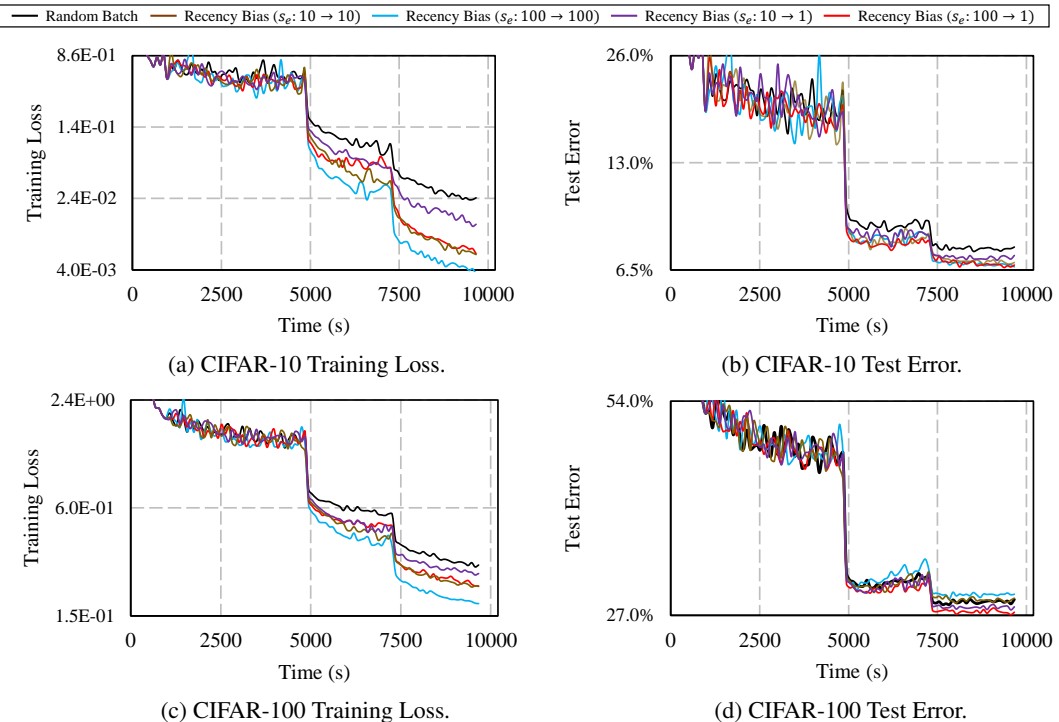

Figure 6: Ablation study on the effect of the selection pressure.

that of *Random Batch* in CIFAR-100 dataset. On the other hand, the two strategies with decaying converged faster than *Random Batch* in both training loss and test error in all datasets because they exploited more diverse training samples by exponentially decaying the selection pressure. Thus, these observations empirically prove that decaying the selection pressure is an effective way to alleviate the overfitting problem at the later stage of training.

## 6 CONCLUSION

In this paper, we presented a novel adaptive batch selection algorithm called *Recency Bias* that emphasizes predictively uncertain samples for accelerating the training of neural networks. Toward this goal, the predictive uncertainty of each sample is evaluated using its *recent* label predictions managed by a sliding window of a fixed size. Then, uncertain samples *at the moment* are selected with high probability for the next mini-batch. We conducted extensive experiments on both classification and fine-tuning tasks. The results showed that *Recency Bias* is effective in reducing the training time as well as the best test error. It was worthwhile to note that using *all* historical observations to estimate the uncertainty has the side effect of slowing down the training process. Overall, a merger of uncertain samples and sliding windows greatly improves the power of adaptive batch selection.

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

## A    HYPERPARAMETER SELECTION

*Recency Bias* receives the two hyperparameters: *(i)* the initial selection pressure $s_{e_0}$ that determines the sampling probability gap between the most and the least uncertain samples and *(ii)* the window size $q$ that determines how many recent label predictions are involved in predicting the uncertainty. To decide the best hyperparameters, we trained ResNet (L=50) on CIFAR-10 and CIFAR-100 with a momentum optimizer. For hyperparameters selection, the two hyperparameters were chosen in a grid $s_{e_0} \in \{1, 10, 100, 1000\}$ and $q \in \{5, 10, 15\}$.

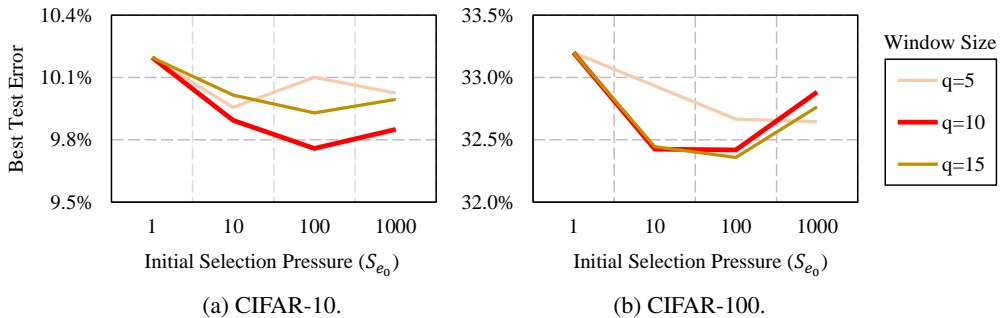

(a) CIFAR-10.                    (b) CIFAR-100.

Figure 7: Grid search on CIFAR-10 and CIFAR-100 datasets using ResNet.

Figure 7 shows the test errors of *Recency Bias* obtained by the grid search on the two datasets. Regarding the initial selection pressure $s_{e_0}$, the lowest test error was typically achieved when the $s_{e_0}$ value was 100. As for the window size $q$, the test error was almost always the lowest when the $q$ value was 10. Similar trends were observed for the other combinations of a neural network and an optimizer. Therefore, in all experiments, we set $s_{e_0}$ to be 100 and $q$ to be 10.

## B    EXPERIMENT USING TINY-IMAGENET DATASET

For a larger-scale experiment, we repeated the image classification task on Tiny-ImageNet (200 classes), a subset of ImageNet (Krizhevsky et al., 2012), with $100,000$ training and $10,000$ validation images. Because no test set exists, we used the validation set as the test data. For Tiny-ImageNet dataset, we trained the network for $80,000$ iterations and used an initial learning rate of $0.1$, which was divided by 10 at $50\%$ and $75\%$ of the total number of training iterations. The remaining experimental configurations were the same as those in Section 5.2.

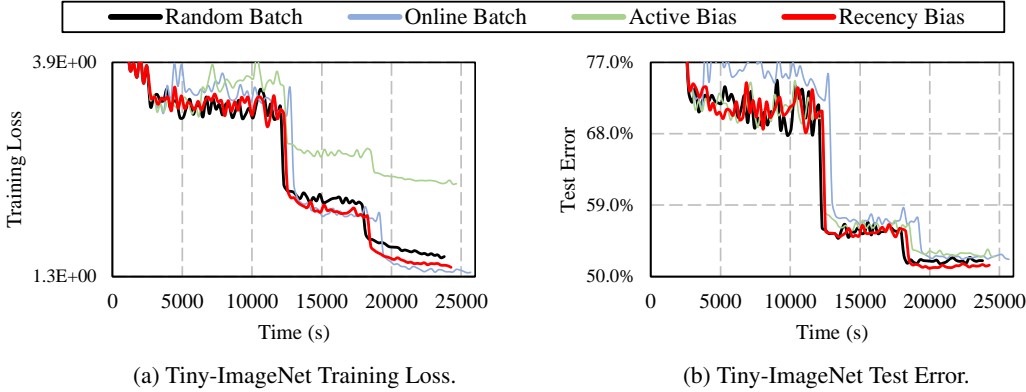

(a) Tiny-ImageNet Training Loss.              (b) Tiny-ImageNet Test Error.

Figure 8: Convergence curves of four batch selection strategies using **DenseNet with momentum**.

Table 4: The best test errors (%) of four batch selection strategies using **DenseNet**.

| Method | *Random Batch* | *Online Batch* | *Active Bias* | *Recency Bias* |
|---|---|---|---|---|
| Tiny-ImageNet | $51.6 \pm 0.26$ | $52.5 \pm 0.19$ | $52.2 \pm 0.52$ | $51.0 \pm 0.34$ |

Figure 8 shows the convergence curves of training loss and test error using four batch selection strategies on Tiny-ImageNet, where the best test errors are detailed in Table 4. Again, only *Recency Bias* converged faster than *Random Batch* in both training loss and test error. On the other hand, although *Online Batch* showed the fastest convergence in training loss, its test error was worse than that of *Random Batch* because of the overfitting to hard training samples. Similarly, the test error of *Active Bias* was also worse than that of *Random Batch* because of the side effect of slowing down the convergence speed of training. In summary, *Recency Bias* achieved the test error relatively lower by 1.16% than *Random Batch*, 2.86% than *Online Batch*, and 2.30% than *Active Bias*.

## C    GENERALIZATION OF *Recency Bias*

### C.1    CONVERGENCE CURVES USING DENSENET WITH SGD

Figure 9 shows the convergence curves of training loss and test error for four batch selection strategies using DenseNet and an SGD optimizer, which corresponds to the right side of Table 1.

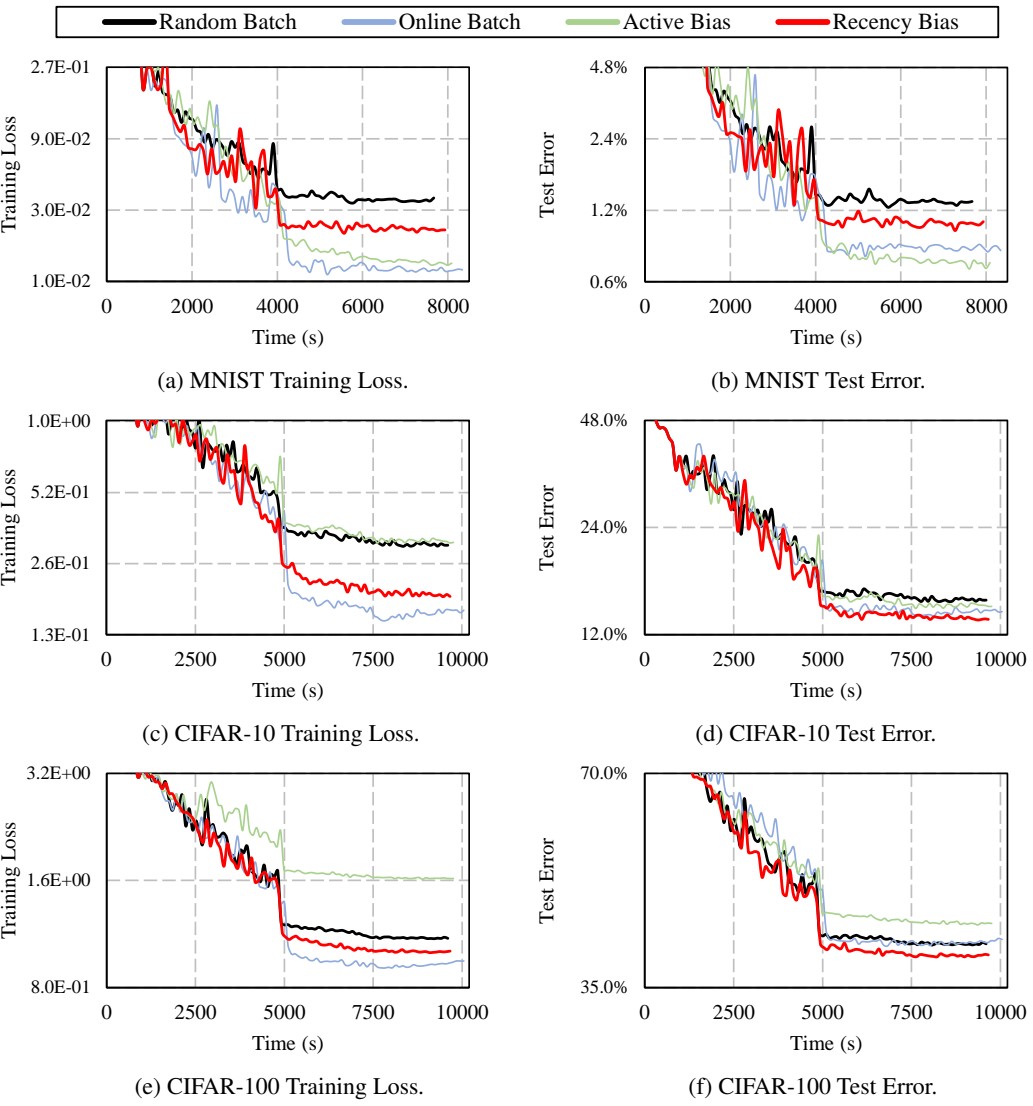

(a) MNIST Training Loss.

(b) MNIST Test Error.

(c) CIFAR-10 Training Loss.

(d) CIFAR-10 Test Error.

(e) CIFAR-100 Training Loss.

(f) CIFAR-100 Test Error.

Figure 9: Convergence curves of four batch selection strategies using **DenseNet with SGD**.

## C.2 CONVERGENCE CURVES USING RESNET WITH MOMENTUM

Figure 10 shows the convergence curves of training loss and test error for four batch selection strategies using ResNet and a momentum optimizer, which corresponds to the left side of Table 2.

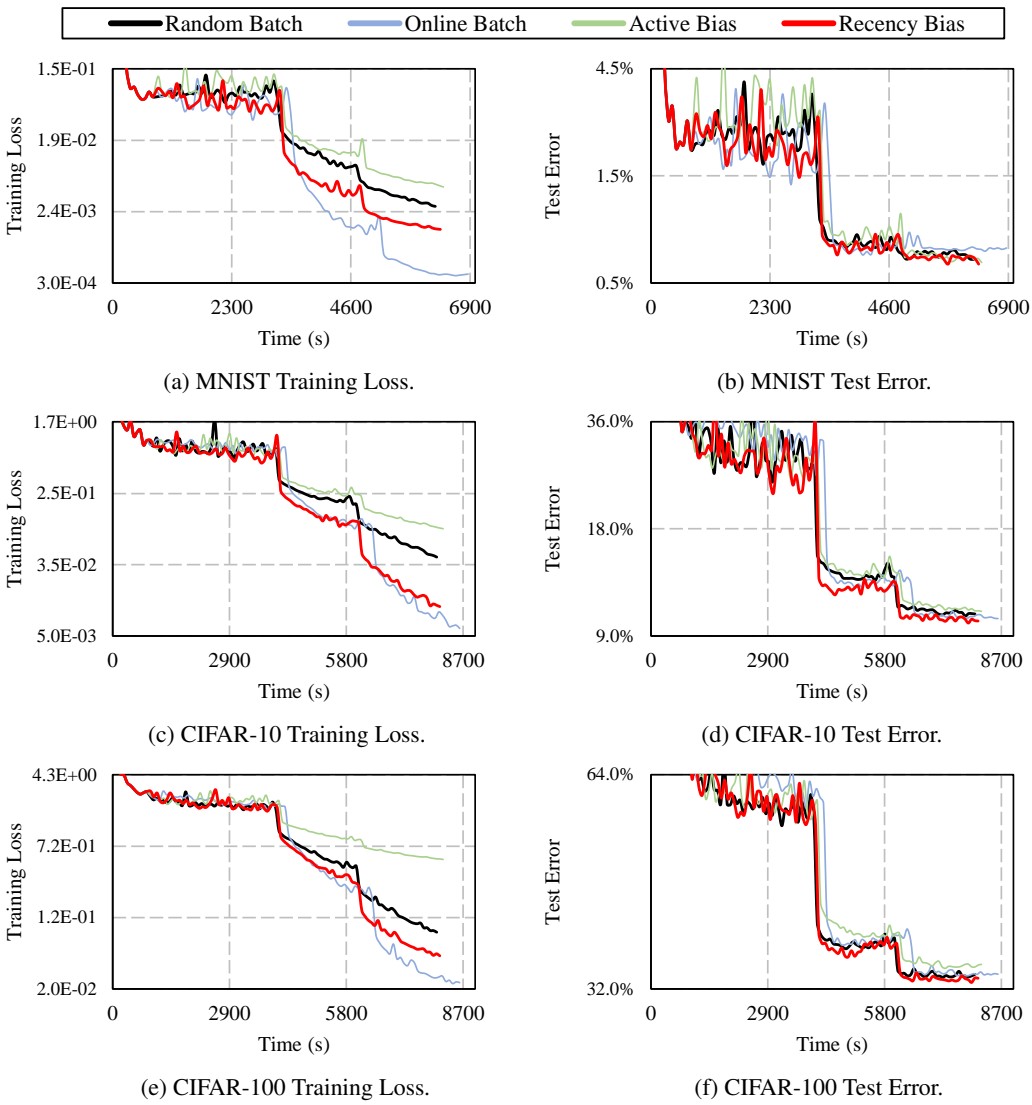

(a) MNIST Training Loss.

(b) MNIST Test Error.

(c) CIFAR-10 Training Loss.

(d) CIFAR-10 Test Error.

(e) CIFAR-100 Training Loss.

(f) CIFAR-100 Test Error.

Figure 10: Convergence curves of four batch selection strategies using **ResNet with momentum**.

## C.3 CONVERGENCE CURVES USING RESNET WITH SGD

Figure 11 shows the convergence curves of training loss and test error for four batch selection strategies using ResNet and an SGD optimizer, which corresponds to the right side of Table 2.

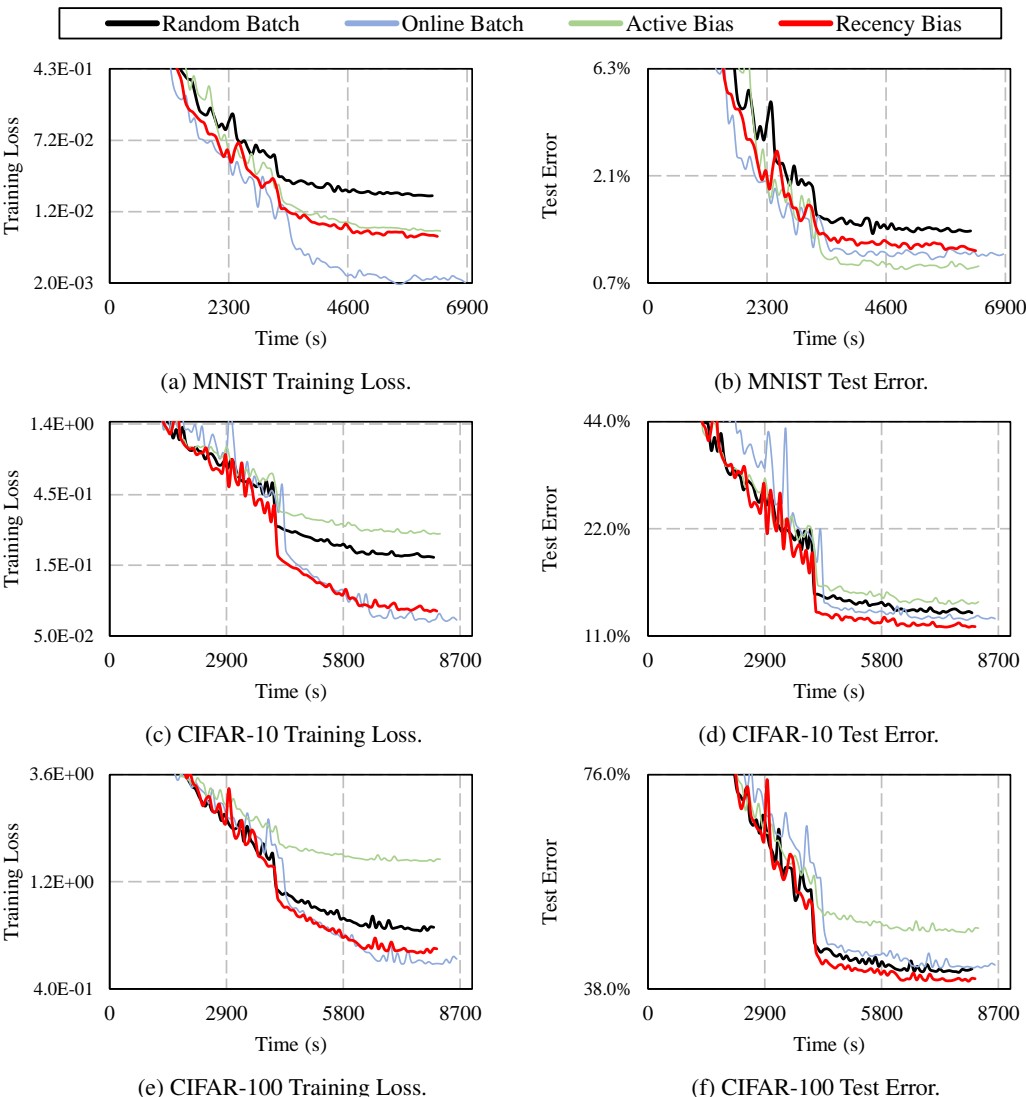

Figure 11: Convergence curves of four batch selection strategies using **ResNet with SGD**.

