# OpenReview forum: "Carpe Diem, Seize the Samples Uncertain "at the Moment" for Adaptive Batch Selection"
_ICLR.cc/2020/Conference — Reject_

### Official Review · AnonReviewer3 · 2019-10-16
**Official Blind Review #3**

**Rating:** 6

**Review:**

This paper explores a well motivated but very heuristic idea for selecting the next samples to train on for training deep learning models. This method relies on looking at the uncertainty of predictions of in the recent history of statements and preferring those instances that have a predictive uncertainty over the recent predictions.  This allows the training method to train on instances that are neither too hard nor too easy and focus on reducing the uncertainty whenever it has the greatest potential gain to do so.

There are two extra components that make this method work:
- Windowing: only looking at the recent history of the instances which has two effects: firstly, the current state of the model is explored which gives a more recent assessment relative to the current state of the model. Secondly, it makes the algorithm faster by reducing the overhead of analyzing the prediction history of samples.
- Annealing the selection bias: as the training goes on the selection becomes more random and less biased.

This approach is evaluated in on three simple data-sets: MNIST, CIFAR-10 and CIFAR-100. Although this is a very limited subset of models, the results are consistent and statistically significant, although their effect is not really huge.

The paper gives very little theoretical justification or analysis of the results but gives only the presented empirical evidence which seems to support the hypothesis on the efficacy of the approach.

Another drawback of the approach is that it introduces new hyperparameters: those governing the annealing schedule for the selection bias.

Since the approach seems efficient in a relatively constrained setup, it can be reasonably expected that it might be helpful in more general situations, therefore. On the other hand, since it is only evaluated on three very similar tasks, it limits the conclusiveness of the results.

That's why I would for weak accept. In the presence of more empirical (or even theoretical) evidence, I would vote for strong accept.



**Experience Assessment:**

I do not know much about this area.

**Review Assessment: Checking Correctness Of Derivations And Theory:**

I assessed the sensibility of the derivations and theory.

**Review Assessment: Checking Correctness Of Experiments:**

I carefully checked the experiments.

**Review Assessment: Thoroughness In Paper Reading:**

I read the paper thoroughly.

---

> ### Author Response · Authors · 2019-11-15
> **Response to Reviewer 3**
>
> Thank you for raising your insightful and detailed comments. We have revised our paper to address your concerns; please see the modified parts colored by “blue”.
>
> Below is our response to your concerns:
>
> Q3-1. Another drawback of the approach is that it introduces new hyperparameters.
> A3-1. According to our experience, q=10 and se=100, in general, worked well in many scenarios. Thus, tuning these hyperparameters was not that tricky.
>
> Q3-2. The approach is only evaluated on three very similar tasks.
> A3-2. This is a very good point. We could not add more tasks during the rebuttal period. We will definitely add more tasks such as text classification to the camera-ready version, if accepted.
>
> Q3-3. In the presence of more empirical (or even theoretical) evidence, I would vote for strong accept.
> A3-3. For more empirical evidence, we have added an ablation study of the effect of the selection pressure (See Section 5.4 for details).  Furthermore, we have included the results for Tiny-ImageNet (See Appendix B for details). Recency Bias outperformed the other methods by 1.16%-2.86%. Overall, the superiority of Recency Bias is further consolidated.

---

### Official Review · AnonReviewer1 · 2019-10-23
**Official Blind Review #1**

**Rating:** 3

**Review:**

This paper proposes an interesting heuristic of batch construction from samples. Instead of the usual random sampling, the authors to sample based on some measures of the ``uncertainty”. To be specific, the uncertainty is measured as a normalized entropy estimated from a window of historical predictions.

I like the idea of designing more sophisticated ways to encourage more exploration over the samples that the model is not good at. The thought is similar as active learning. It is interesting to see how similar thought can be used to improve the performance of the algorithm in the general batch gradient descent setting.

On the other hand I am not quite convinced the proposed way is truly better. The main concern is the experiments do not quite show the state-of-the-art result at all. It is not even close on MNIST, CIFAR-10 and CIFAR-100. Also those datasets are relatively small one. Can authors add results on larger datasets such as tiny image net?

Besides this main concern I also have some worries about the design of the algorithm. I listed them below:
1. The vanilla stochastic gradient descent can be roughly justified since the expectation of the stochastic gradient is the true gradient of the loss. Now with the proposed heuristic will this still be true?
2. Is there any guarantee the algorithm can converge? It is not clear to me as the optimization proceeds the ``uncertainty” may oscillate. Is there any condition when the convergence is guaranteed?
3. As the number of classes grows the estimation of the entropy itself is a tough problem. Is there any way to mitigate this issue other than increase the window size?

Another minor comment:
Could the authors add more explanation on equation (4)? For example, $\delta$ is related to the maximum entropy led by a uniform distribution, and the summation term in (4) is related to the empirical entropy.



**Experience Assessment:**

I have read many papers in this area.

**Review Assessment: Checking Correctness Of Derivations And Theory:**

I assessed the sensibility of the derivations and theory.

**Review Assessment: Checking Correctness Of Experiments:**

I assessed the sensibility of the experiments.

**Review Assessment: Thoroughness In Paper Reading:**

I read the paper at least twice and used my best judgement in assessing the paper.

---

> ### Author Response · Authors · 2019-11-15
> **Response to Reviewer 1**
>
> Thank you for raising your insightful and detailed comments. We have revised our paper to address your concerns; please see the modified parts colored by “blue”.
>
> Below is our response to your concerns:
>
> Q1-1. The experiments do not quite show the state-of-the-art result at all.
> A1-1. Yes, this is a painful point for us. The state-of-the-art performance is usually obtained by a very complex network which requires very high computing resources (e.g., may GPUs). Most university labs do not equip such computing resources, and many relevant papers from universities, in fact, have the same issue and do not show the state-of-the-art performance. This unavoidable limitation needs to be considered. Nevertheless, we expect that our technique will be effective also in a very complex network.
>
> Q1-2. The authors can add results on larger data sets such as Tiny-ImageNet.
> A1-2. Thank you for the suggestion. Yes, we have added the results for Tiny-ImageNet, and Recency Bias outperformed the other methods by 1.16%-2.86%. Please see Appendix B of the updated paper.
>
> We could not complete the experiment on a variety of networks (or optimizers) because each run took 25,000 seconds. But, we will cover all the results to the camera-ready version, if accepted.
>
>
> Q1-3. It is not clear how the proposed heuristic is justified.
> A1-3. The below paper[1] proved that preferring confusable(uncertain) samples not only leads to convergence but also brings larger gradient contribution, which justifies the adaptive batch selection. We will clarify this justification in the camera-ready version, if accepted.
>
> Q1-4. It is not clear whether the algorithm is guaranteed to converge.
> A1-4. Yes, an adaptive batch selection algorithm is guaranteed to converge[1]. Thus, the papers in this topic usually do not repeat formal convergence analysis.
>
> [1] Adaptive sampling for SGD by exploiting side information (ICML’16)
>
> Q1-5. Estimating the entropy with many classes could be tough.
> A1-5. According to our experience, even though there were many classes (i.e., CIFAR-100), prediction results for the same sample were confused among only a few class labels. Thus, many classes did not cause a problem.
>
> Q1-6. Please add more explanation on Eq. (4).
> A1-7. Yes, we have added more explanation on Eq. (4) in Section 3.

---

### Official Review · AnonReviewer2 · 2019-10-26
**Official Blind Review #2**

**Rating:** 3

**Review:**

This paper proposes Recency Bias, an adaptive mini batch selection method for training deep neural networks. To select informative minibatches for training, the proposed method maintains a fixed size sliding window of past model predictions for each data sample. At a given iteration, samples which have highly inconsistent predictions within the sliding window are added to the minibatch. The main contribution of this paper is the introduction of sliding window to remember past model predictions, as an improvement over the SOTA approach: Active Bias, which maintains a growing window of model predictions. Empirical studies are performed to show the superiority of Recency Bias over two SOTA  approaches. Results are shown on the task of (1) image classification from scratch and (2) image classification by fine-tuning pretrained networks.

+ves:
+ The idea of using a sliding window over a growing window in active batch selection is  interesting.
+ Overall, the paper is well written. In particular, the Related Work section has a nice flow and puts the proposed method into context. Despite the method having limited novelty (sliding window instead of a growing window), the method has been well motivated by pointing out the limitations in SOTA methods.
+ The results section is well structured. It's nice to see hyperparameter tuning results; and loss convergence graphs in various learning settings for each dataset.

Concerns:
- The key concern about the paper is the lack of rigorous experimentation to study the usefulness of the proposed method. Despite the paper stating that there have been earlier work (Joseph et al, 2019 and Wang et al, 2019) that attempt mini-batch selection, the paper does not compare with them. This is limiting. Further, since the proposed method is not specific to the domain of images, evaluating it on tasks other than image classification, such as text classification for instance, would have helped validate its applicability across domains.

- Considering the limited results, a deeper analysis of the proposed method would have been nice. The idea of a sliding window over a growing window is a generic one, and there have been many efforts to theoretically analyze active learning over the last two decades. How does the proposed method fit in there? (For e.g., how does the expected model variance change in this setting?) Some form of theoretical/analytical reasoning behind the effectiveness of recency bias (which is missing) would provide greater insights to the community and facilitate further research in this direction.

- The claim of 20.5% reduction in test error mentioned in the abstract has not been clearly addressed and pointed out in the results section of the paper.

- On the same note, the results are not conclusively in favor of the proposed method, and only is marginally better than the competitors. Why does online batch perform consistently than the proposed method? There is no discussion of these 	inferences from the results.

- The results would have been more complete if results were shown in a setting where just recency bias is used without the use of the selection pressure parameter. In other words, an ablation study on the effect of the selection pressure parameter would have been very useful.

- How important is the warm-up phase to the proposed method? Considering the paper states that this is required to get good estimates of the quantization index of the samples, some ablation studies on reducing/increasing the warm-up phase and showing the results would have been useful to understand this.

- Fig 4: Why are there sharp dips periodically in all the graphs? What do these correspond to?

- The intuition behind the method is described well, however, the proposed method would have been really solidified if it were analysed in the context of a simple machine learning problem (such as logistic regression). As an example, verifying if the chosen minibatch samples are actually close to the decision boundary of a model (even if the model is very simple) would have helped analyze the proposed method well.

Minor comments:
* It would have been nice to see the relation between the effect of using recency bias and the difficulty of the task/dataset.
* In the 2nd line in Introduction, it should be "deep networks" instead of "deep networks netowrks".
* Since both tasks in the experiments are about image classification, it would be a little misleading to present them as "image classification" and "finetuning". A more informative way of titling them would be "image classification from scratch" and "image classification by finetuning".
* In Section 3.1, in the LHS of equation 3, it would be appropriate to use P(y_i/x_i; q) instead of P(y/x_i; q) since the former term was used in the paragraph.

=====POST-REBUTTAL COMMENTS========
I thank the authors for the response and the efforts in the updated draft. Some of my queries were clarified. However, unfortunately, I still think more needs to be done to explain the consistency of the results and to study the generalizability of this work across datasets.  I retain my original decision for these reasons.

**Experience Assessment:**

I have published in this field for several years.

**Review Assessment: Checking Correctness Of Derivations And Theory:**

I assessed the sensibility of the derivations and theory.

**Review Assessment: Checking Correctness Of Experiments:**

I carefully checked the experiments.

**Review Assessment: Thoroughness In Paper Reading:**

I read the paper at least twice and used my best judgement in assessing the paper.

---

> ### Author Response · Authors · 2019-11-15
> **Response to Reviewer 2**
>
> Thank you for raising your insightful and detailed comments. We have revised our paper to address your concerns; please see the modified parts colored by “blue.”
>
> Below is our response to your concerns:
>
> Q2-1. Evaluating the proposed method on other tasks (e.g., text classification) would have helped validate its applicability across domains.
> A2-1. This is a very good point. We could not add more tasks during the rebuttal period. We will definitely add more tasks such as text classification to the camera-ready version, if accepted.
>
> Q2-2. The claim of 20.5% reduction in test error mentioned in the abstract has not been clearly addressed and pointed out in the result section of the paper.
> A2-2. 20.5% was obtained from Table 1 (Momentum - MNIST). |0.616 (Active Bias) – 0.490 (Recency Bias)| / 0.616 = 20.5%. We will clarify this value in the camera-ready version, if accepted.
>
> Q2-3. The proposed method is marginally better than the competitors.
> A2-3. We understand your concern. Nevertheless, Recency Bias outperformed Online Batch by 2.22-8.86% and Active Bias by 0.98-20.5% in Tables 1 and 2. We believe that these improvements are "not" insignificant, though are not big either.
>
> Q2-4. An ablation study on the effect of the selection pressure parameter would have been very useful.
> Q2-4. We have added the ablation study on the effect of the selection pressure. Please see Section 5.4 of the updated paper.
>
> Q2-5. It is not clear how important is the warm-up phase to the proposed method?
> A2-5. The warm-up phase is simply for accumulating the prediction history, which is needed to calculate predictive uncertainty in Definition 3.1, up to the window size q. Hence, it is not a target for ablation studies.
>
> Q2-6. Why are there sharp dips periodically in all the graphs?
> A2-6. The sharp dip is caused by the learning rate decay, which is commonly observed in deep learning.
>
> Q2-7. The proposed method would have been really solidified if it were analyzed in the context of a simple machine learning problem (such as logistic regression).
> A2-7. Thank you for your suggestions. We understand your point, but, since our main goal is to train a deep neural network, analysis in the context of a simple machine learning algorithm may not hold for a deep neural network. However, we fully agree that more in-depth analysis should solidify Recency Bias.
>
> Furthermore, for more empirical evidence, we have included the results for Tiny-ImageNet. Recency Bias outperformed the other methods by 1.16%-2.86%. Please see Appendix B of the updated paper.

---

### Decision · Program_Chairs · 2019-12-19

**Decision:**

Reject

**Comment:**

The authors propose a new mini-batch selection method for training deep NNs. Rather than random sampling, selection is based on a sliding window of past model predictions for each sample and uncertainty about those samples. Results are presented on MNIST and CIFAR.

The reviewers agreed that this is an interesting idea which was clearly presented, but had concerns about the strength of the experimental results, which showed only a modest benefit on relatively simple datasets. In the rebuttal period, the authors added an ablation study and additional results on Tiny-ImageNet. However, the results on the new dataset seem very marginal, and R1 did not feel that all of their concerns were addressed. I’m inclined to agree that more work is required to prove the generalizability of this approach before it’s suitable for acceptance.